# New gridded dataset of rainfall erosivity (1950–2020) on the Tibetan Plateau

Yueli Chen[1], Xingwu Duan[2], Minghu Ding[1], Wei Qi[1], Ting Wei[1], Jianduo Li[3,1], Yun Xie[4]

[1]State Key Laboratory of Severe Weather, Chinese Academy of Meteorological Sciences, Beijing, 100081, China

[2]Institute of International Rivers and Eco-security, Yunnan University, Kunming, 650091, China

[3]CMA Earth System Modeling and Prediction Centre, Beijing, 100081, China

[4]State Key Laboratory of Earth Surface Processes and Resources Ecology, Faculty of Geographic
Science, Beijing Normal University, Beijing, 100875, China

*Correspondence to:* Xingwu Duan (xwduan@ynu.edu.cn), Minghu Ding (dingminghu@foxmail.com)

**Abstract.** The risk of water erosion on the Tibetan Plateau (TP), a typical fragile ecological area, is increasing with climate change. Rainfall erosivity map is useful for understanding the spatial-temporal

pattern of rainfall erosivity and identifying hot spots of soil erosion. This study generated an annual gridded rainfall erosivity dataset on a 0.25° grid for the TP in 1950–2020. 1-min precipitation observations at 1787 weather stations for 7 years and 0.25° hourly European Center for Medium-Range Weather Forecasts Reanalysis 5 (ERA5) precipitation data for 71 years are employed in this study. Our results indicated that the ERA5-based estimates have a marked tendency to underestimate annual

rainfall erosivity when compared to the station-based estimates, because of the systematically biases of ERA5 precipitation data including the large underestimation of the maximum contiguous 30-min peak intensity and relatively slight overestimation of event erosive precipitation amount. The multiplier factor map over the TP, which was generated by Inverse Distance Weighted method based on the relative changes between the available station-based annual rainfall erosivity grid values and the

corresponding ERA5-based values, was employed to correct the ERA5-based annual rainfall erosivity and then reconstruct the annual rainfall erosivity dataset. The multi-year averaged correction coefficient over the TP between the station-based annual rainfall erosivity values and the newly released data is 0.67. In addition, the probability density and various quantile values of the new data are generally consistent with the station-based values. The data offers a view of large-scale spatial-temporal

variability in the rainfall erosivity and addresses the growing need for the information to predict

rainfall-induced hazards over the TP. The dataset are available at

http://data.tpdc.ac.cn/en/data/37c34046-3c2a-4737-b3c9-35af398da62a/.

## 1 Introduction

Precipitation is the main driver of water erosion because it directly affects the detachment of soil particles, breakdown of aggregates, and transport of eroded particles via runoff (Wischmeier and Smith, 1965, 1978). The $R$ factor, that is, the multi-year average rainfall erosivity, which is described by the Universal Soil Loss Equation (USLE; Wischmeier and Smith, 1965, 1978) and Revised USLE (RUSLE; Renard, 1997), is an indicator of the multi-year average potential ability of rainfall and runoff to affect soil erosion. The $R$ factor is calculated using the classical (Wischmeier and Smith, 1965) and statistical algorithms (e.g., Liu et al., 2002) according to the temporal resolution of the precipitation data.

The classical algorithm for rainfall erosivity requires a continuous precipitation data series with <15-min temporal resolution (Angulo-Martínez and Beguería, 2009). As networks of weather stations and observation platforms have matured considerably in the past two decades, rainfall erosivity has been calculated using the classical algorithm at the local scale (Agnese et al., 2006; Ma et al., 2014; Wang et al., 2017), and the application of the algorithm has been gradually extended to the national (Panagos et al., 2015; Kim et al., 2020; Yue et al., 2021) and global scale (Panagos et al., 2017; Liu et al., 2020). Despite substantial progress, it is still notable that the relative error of the estimated rainfall erosivity increases rapidly with increasing time interval of the precipitation data. For example, the relative error based on hourly data was more than 80%, compared with the results based on 1-min data (Lobo and Bonilla, 2015; Yin et al., 2015; Shin et al., 2019). In addition, the accuracy of the rainfall erosivity is greatly reduced by inadequate weather station coverage, especially in areas with complex climates and terrains (Yue et al., 2021). Therefore, the accuracy of rainfall erosivity estimation depends strongly on the temporal and spatial resolution of the precipitation observations (Panagos et al., 2017; Kim et al., 2020).

Compared with in-situ observations, gridded precipitation data (e.g. satellite-based, reanalysis and fused datasets) are not subjected to topographical limitations and could supply continuous precipitation data (Beck et al., 2017). These data have been widely used to estimate the rainfall erosivity in China, especially in the regions with scarce in-situ observations (Teng et al., 2018), Germany (Risal et al., 2018), Africa (Vrieling et al., 2010), the United States (Kim et al., 2020), and other regions. They have contributed greatly to our knowledge of the spatiotemporal patterns of rainfall erosivity; however, the uncertainties in rainfall erosivity directly calculated by using gridded precipitation data have not been quantified, although obvious biases between gridded and observed precipitation values have been

demonstrated (Freitas et al., 2020).

The Tibetan Plateau (TP) referred to as the Third Pole is one of the highest plateaus worldwide and has an average altitude of more than 4000 m (Yao et al., 2012). Since the mid-1950s, the TP has experienced significant warming exceeding that of other regions in the same latitude zone (Liu and Chen, 2000). Owing to increasing snowmelt and more frequent heavy precipitation events, which may cause more severe soil erosion, knowledge of the rainfall erosivity on the TP is highly important for soil sustainability and thus water and food security. The accuracy of rainfall erosivity estimation depends mainly on the spatiotemporal accuracy of the precipitation data, especially in the TP, where the seasonal and regional precipitation patterns exhibit significant variability owing to westerly winds, the Indian monsoon, and land–atmosphere interaction.

Many efforts have been made to study the rainfall erosivity on the TP (Table 1). Most studies employed the empirical methods, however, our study has demonstrated that these empirical methods always resulted to obvious biases over the TP, when compared with the results based on the 1-min precipitation data by using the standard method (paper submitted). In the term of the type of the precipitation data, dozens of station-based precipitation data were commonly used to calculate the rainfall erosivity (e.g., Qin et al., 2016; Gu et al., 2020). Yue et al. (2022) has reported that the scarce weather stations can significantly reduce the estimation accuracy of the rainfall erosivity in the regions with complex terrain and climate, especially in the TP. Therefore, the accuracy of the estimated rainfall erosivity in the TP are largely reduced by the current empirical estimation models and the scarcity of the historical weather stations. In other words, the precipitation data with high spatial-temporal resolution are essential to calculate the rainfall erosivity in the TP.

**Table 1.** *R* factor of TP in previous studies

| Study scale | Type of precipitation data | Number of weather stations | Temporal resolution | Calculation Method | Spatial characteristics | *R* factor ($MJ \cdot mm \cdot ha^{-1} \cdot h^{-1} \cdot yr^{-1}$) | Reference |
|---|---|---|---|---|---|---|---|
| China | Weather station | China: 2381<br>TP: < 100 | 1 hour | Standard | Kriging | TP: 273 | Yue et al., 2022 |
| Southwestern China | CRU_TS4 | | Monthly | Empirical | Grid,<br>no bias correction | Tibet: 3407 | Cao et al., 2018 |
| China | TRMM,<br>weather station | China: 650<br>TP: < 50 | Daily | Empirical | Grid,<br>bias correction of TRMM precipitation | No value | Teng et al., 2017 |
| China | Weather station | China: 756<br>TP: < 50 | Daily | Empirical | Kriging | TP: 408 | Qin et al., 2016 |
| China's dryland region | Weather station | China's dryland region 298 | Daily | Empirical | HASM interpolation | Most of TP: 1–500 | Yang et al., 2015 |
| China | Weather station | China: 590<br>TP: < 50 | Daily | Empirical | Kriging | Central and eastern TP: 147 | Liu et al., 2013 |
| China | Weather station | China: 564 | Daily | Empirical | Kriging | Cold zone of TP: 368<br>Sub-cold zone of TP: 427 | Zhang et al., 2003 |
| Tibet | Weather station | Tibet: 38 | Daily | Empirical | Station-averaged | Tibet: 714 | Gu et al., 2020 |
| Tibet | TRMM 3B42 gridded | | Daily | Empirical | Grid,<br>no bias correction | Tibet: 768 | Yan et al., 2010 |

Note: CRU_TS4: Climatic Research Unit Time Series 4 gridded precipitation product. TRMM: Tropical Rainfall Measuring Mission gridded precipitation product. Empirical method means the rainfall erosivity values are calculated by using the empirical equations based on daily or monthly precipitation data. The standard method is proposed by USLE or RUSLE. The boundaries of the TP used in these studies has some slight differences.

To expand the spatial coverage and extend the time series of rainfall erosivity over the TP, the various gridded precipitation datasets, for example, satellite-based Tropical Rainfall Measuring Mission (TRMM) and station-based Climatic Research Unit Time Series 4 gridded precipitation datasets (CRU_TS4), are also introduced into the soil erosion study of the TP in recent decade (Yan et al., 2010; Teng et al., 2017; Gao et al., 2018). The performances of these gridded precipitation mainly depend on the spatial-temporal accuracy of the gauge observations, and thus these datasets always present obvious biases, due to insufficient density of the weather station network over the TP (Yuan et al., 2021). It is notable that the significant biases of various gridded precipitation data have also been widely identified (Sun et al., 2018), and their impacts on the rainfall erosivity estimation have not been assessed.

In recent researches, the model-based gridded precipitation datasets begin to be concerned (Li et al., 2020; Zhou et al., 2021), because they could resolve the complex topography and climate effects over the TP and provide long-term data by setting simulation period. The European Center for Medium-Range Weather Forecasts Reanalysis 5 (ERA5) as the newly generation is one of the most widely used precipitation datasets in the world (Hersbach et al. 2019). Compared with other gridded precipitation datasets, ERA5 succeeded in reproducing the inter-annual and decadal variabilities of precipitation and reflecting the spatial-temporal patterns (Yuan et al., 2021), and performed marginally better in detecting daily precipitation over the whole TP for the long-term periods (Jiang et al., 2021), despite the bias in precipitation amount was also reported (Jiang et al., 2021; Jiao et al., 2021). Therefore, this study aims to reconstruct the historical annual rainfall erosivity with 0.25° spatial resolution in 1950–2020 over the TP, by employing the 0.25° hourly ERA5 precipitation data for 71 years to generate a long-term background values and utilizing the 1-min precipitation observations at 1787 weather stations for 7 years to identify and correct the biases of the estimates. In detail, this paper describes (1) the performance of ERA5 precipitation data at the weather stations; (2) the performance of the ERA5-based annual rainfall erosivity calculated by using the standard method recommended by the USLE model; (3) the correction of the ERA5-based annual rainfall erosivity and the validation of the newly generated dataset.

## 2 Study Area and Source Data

### 2.1 Tibetan Plateau

The study area is the TP (26–40°N, 73–105°E), which is located in Southwestern China and covers an

area of approximately 2.5 million km$^2$. The elevation of the TP ranges from 84 to 8246 m, with an average value of 4379 m. Precipitation in the southeastern TP is influenced by warm, humid Indian monsoons, whereas in the western TP, it is influenced more strongly by the mid-latitude westerlies (Yao et al., 2012). The annual precipitation is concentrated from May to October (Gu et al., 2020), and shows a spatial pattern of a wet east and west with a dry middle (Li et al., 2020). Along with the significant climate change and a very fragile ecological environment, the TP has high potential for soil loss, especially in the eastern TP and Hengduan Mountains, which are among the most severely eroded areas in China (Teng et al., 2019).

**2.2 Precipitation data**

Previous studies of the TP have used in-situ precipitation observations with <50 stations and coarse temporal resolution, e.g., hourly (Yue et al., 2021), daily (Wang et al., 2017), or half-monthly (Teng et al., 2018; Gu et al., 2020; Liu et al., 2020). By contrast, this study estimated the rainfall erosivity on the TP using precipitation observations at 1-min intervals in 2013–2020 at 1787 weather stations obtained from the National Meteorology Information Center of the China Meteorological Administration [Figure 1(a)].

To ensure the accuracy of the in situ precipitation data, we evaluated their quality. The data integrity of each station was first checked using quality control codes at 1-min intervals by month. Because precipitation on the TP occurs mainly from May to September, observed data with an integrity of >90% from May to September in a year can be used to calculate the annual rainfall at the station. The number of stations with data suitable for calculating the annual rainfall erosivity for each year is shown in the lower left corner of Figure 1(a); it ranges from 628 to 1472, with an average of 1114 stations for 2013–2020 (excluding 2017, because a disruption in data reception caused the loss of precipitation observations in August 2017). Moreover, we examined the station density in each 0.25° grid, which is consistent with the spatial resolution of the ERA5 data [Figure 1(b)]. The number of stations in each grid varies from 1 to 29, and the mean value is 2.1. A total of 836 grids (20% of the grids covering the TP) have observed precipitation values. Because the data quality varies, the available grids with observations change annually; on average, there are 589 available grids with observation records for 2013–2020, excluding 2017.

The hourly 0.25° ERA5 data represent the most recent generation of ECMWF global atmospheric

reanalysis and offer higher spatial resolution than ERA-Interim and other improvements since 1979

(Hersbach et al., 2019). The precipitation data are the sum of large-scale precipitation and convective

precipitation consisting of rain and snow, as determined by the ECMWF Integrated Forecasting

System.

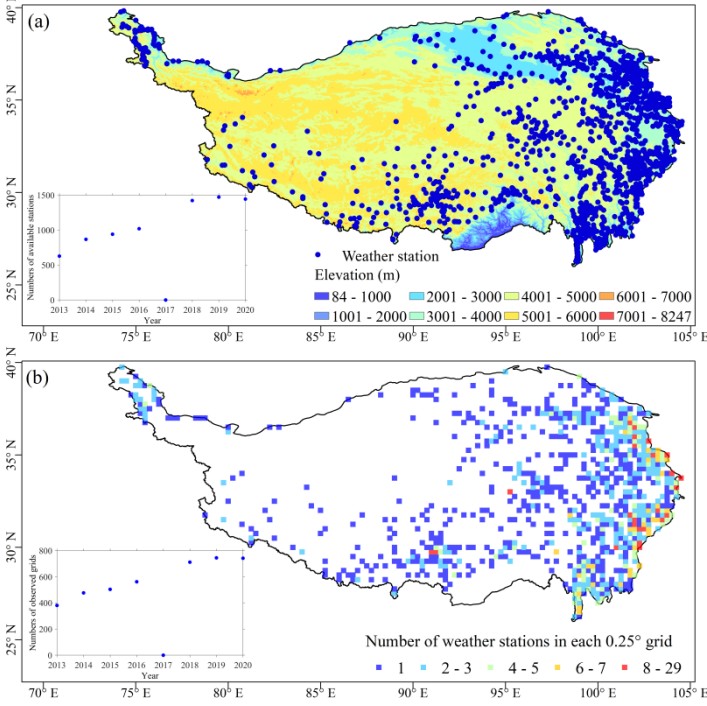

Figure 1. (a) Spatial distribution of weather stations on TP; the inset shows the number of available weather
stations by year. (b) Number of available weather stations in each grid with 0.25° spatial resolution; the inset
shows the number of available weather stations by year.

**3 Methodology**

Figure 2 shows the overall algorithm for generating the annual rainfall erosivity dataset with the 0.25°

spatial resolution over the TP in 1950–2020. We firstly calculated the annual rainfall erosivity by using

the standard method of rainfall erosivity based on the 1-min in-situ precipitation observations and 0.25°

hourly ERA5 precipitation data, respectively. Secondly, the performances of the ERA5 were

systematically assessed in the terms of the detecting accuracy of the precipitation for erosive events and

the estimation accuracy of ERA5-based annual rainfall erosivity. Finally, the historical annual rainfall

erosivity data for the TP was produced after correcting the ERA5-based annual rainfall erosivity.

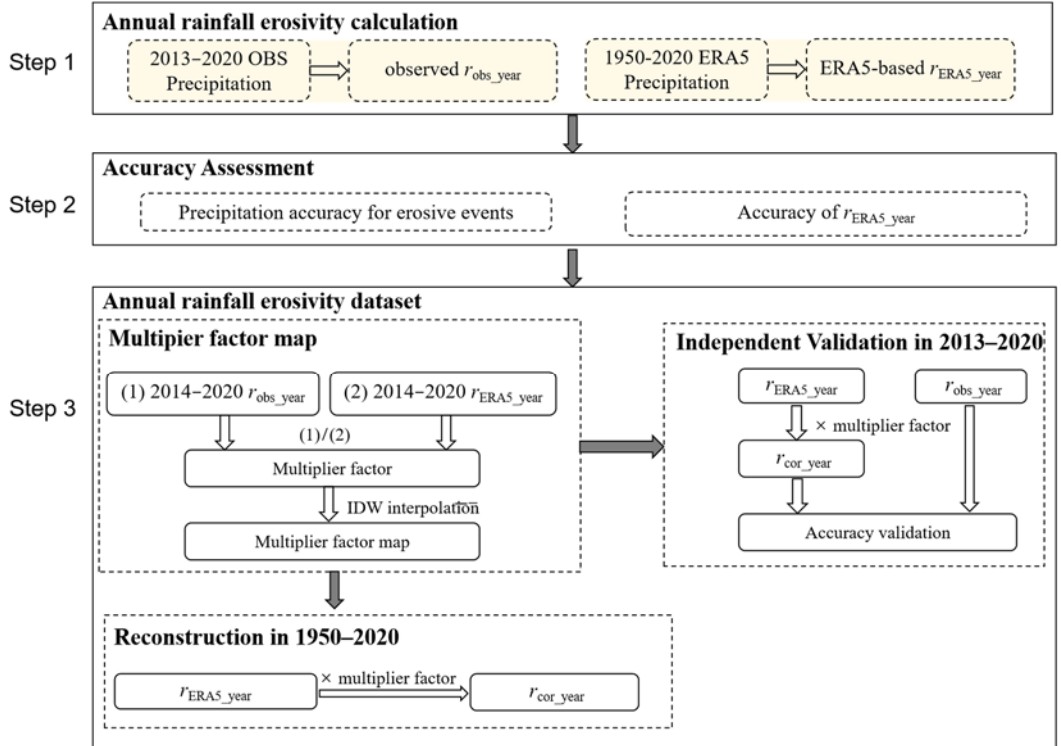

Figure 2. Schematic representation of algorithm for generating annual rainfall erosivity dataset for 1950–2020. $r_{obs\_year}$ and $r_{ERA5\_year}$ represent the station-based and ERA5-based annual rainfall erosivity values, respectively. $r_{cor\_year}$ means the corrected values of the $r_{ERA5\_year}$ by using the multiplier factor map.

## 3.1 Algorithm of annual rainfall erosivity

A rainfall event is defined following Wischmeier and Smith (1978) as having measurable rainfall with no interruption or at most a 6-h interruption. If a rainfall event is interrupted for more than 6 h, subsequent rainfall is considered to belong to a new rainfall event. Rainfall events of more than 12 mm are selected as erosive events following Xie et al. (2000), and the $EI_{30}$ index of the erosive event is calculated. Specifically, the rainfall erosivity of an erosive rainfall event is calculated as follows (Brown and Foster, 1987):

$$e_r = 0.29[1 - 0.72\exp(-0.05i_r)] \tag{1}$$

$$E = \sum_{r=1}^{n}(e_r \cdot P_r) \tag{2}$$

$$r_{event} = E \cdot I_{30} \tag{3}$$

where $E$ (MJ·ha$^{-1}$) is the total energy of the erosive event, and $r_{event}$ (MJ·mm·ha$^{-1}$·h$^{-1}$) is the event rainfall erosivity of the event. For the 1-min precipitation data (ERA5 data), $i_r$ (mm/h) is the rainfall intensity for the $r^{th}$ minute (hour), $e_r$ (MJ·ha$^{-1}$·mm$^{-1}$) is the unit energy for the $r^{th}$ minute (hour), $P_r$

(mm) is the rainfall amount for the $r^{\text{th}}$ minute (hour), $n$ is the rainfall duration, and $I_{30}$ (mm/h) is the maximum contiguous 30-min (1-h) peak intensity. After the event rainfall erosivity at all stations was calculated, we identified and removed extreme outliers of the event rainfall erosivity at each site, which resulted from temporary abnormalities in the automatic observation equipment and were not identified during quality control of the precipitation data. We used boxplots to detect extreme outliers. The lower and upper quartiles were defined as the 25th percentile of event rainfall erosivity (Q1) and the 75th percentile (Q2); the difference (Q2 − Q1) is called the interquartile range (IQR). Event rainfall erosivity data at a station outside the lower and upper bounds (Q1 − 3IQR, Q2 + 3IQR) are considered extreme outliers.

The observed annual rainfall erosivity values ($r_{\text{station\_year}}$) were obtained by summing the rainfall erosivity for all erosive events per year by station. Next, the ERA5-based annual rainfall erosivity ($r_{\text{ERA5\_year}}$) for all the grids in the TP were calculated. Notably, for easy comparison of $r_{\text{station\_year}}$ and $r_{\text{ERA5\_year}}$, the $r_{\text{station\_year}}$ values were upscaled to the grid values ($r_{\text{obs\_year}}$) with 0.25° spatial resolution by averaging the station-based values in the same grid. Figure 1(b) shows the spatial distribution of the available grids with $r_{\text{obs\_year}}$. Steps 2 to 3 in Figure 2 are all based on $r_{\text{obs\_year}}$ and $r_{\text{ERA5\_year}}$ data at grid scale.

**3.2 Assessment of the performance of the ERA5 precipitation data**

The performance of the ERA5 precipitation data were assessed at 280 grid cells, which corresponded to 7% of all the grids over the TP. Given the importance of erosive rainfall events to soil erosion, we focused on the performance of the ERA5 precipitation data in detecting characteristics of erosive rainfall event, including multi-year averaged annual erosive precipitation amount and frequency, and mean erosive event precipitation amount and $I_{30}$.

The mean values of $r_{\text{ERA5\_year}}$ for 2013–2020 were compared with those of $r_{\text{obs\_year}}$ by station. The absolute bias (*AB*) and correction coefficient (*r*) were used to evaluate the accuracy of annual rainfall erosivity estimation using ERA5 data. The *AB* is calculated as shown in Eq. 4.

$$AB = \ \sum_{i=1}^{n}(r_{\text{ERA5\_year}_i} - r_{\text{obs\_year}_i})/n \tag{4}$$

where $i$ is the $i^{\text{th}}$ annual rainfall erosivity value, $r_{\text{ERA5\_year}_i}$ is the ERA5-based annual rainfall erosivity in the $i^{\text{th}}$ year, $r_{\text{obs\_year}_i}$ is the observed annual rainfall erosivity in the $i^{\text{th}}$ year, and $n$ is the number of years of data. Moreover, the empirical orthogonal function (EOF) was employed to assess the

spatiotemporal pattern of annual rainfall erosivity revealed by the ERA5 reanalysis precipitation data by comparing it with the pattern revealed by the observed values.

**3.3 Reconstruction and validation of annual rainfall erosivity**

For the soil erosion process, it is known that not all the precipitation events but the erosive events have close relationship with the water erosion process. Our study indicated the precipitation characteristics derived from ERA5 data for erosive events showed high correction with those from in-situ precipitation observations over the TP (Figure 4). In addition, there was a high correlation between the station-based and ERA5-based annual rainfall erosivity (Figure 5), and their spatiotemporal distribution patterns also

showed well agreement (Figure 7). These findings have demonstrated that it is reasonable to generate the rainfall erosivity dataset for the TP by using the ERA5 precipitation data, and meanwhile, the correction is also essential because of the obvious biases identified in the ERA5-based rainfall erosivity values.

Relative changes between the in-situ and modeled precipitation are always used to correct the modeled

precipitation for accuracy improvement, such as the global precipitation data from WorldClim (Fick et al., 2017), the gridded precipitation data of the China Meteorological Forcing Dataset (He et al., 2020) and the bias adjusted ERA5 precipitation data (Cucchi et al., 2020). Given the close correlation between the precipitation and rainfall erosivity, the relative changes were also employed to correct the ERA5-based annual rainfall erosivity in this study. Here, we have used a hypothesis that the bias of the

ERA5-based annual rainfall erosivity resulted from ERA5 precipitation data at each grid keeps steady by year. In detail, the correction process can be divided into three steps. Firstly, the $r_{\text{obs\_year}}$ values were divided by $r_{\text{ERA5\_year}}$ for each year, and then the calculated results, i.e., the multiplier factor values, were averaged for years. Secondly, inverse distance weighted (IDW) interpolation was used to generate a multiplier factor map of the TP with 0.25° spatial resolution. Thirdly, the corrected annual rainfall

erosivity dataset ($r_{\text{cor\_year}}$) was obtained as the product of $r_{\text{ERA5\_year}}$ and the multiplier factor for each grid.

Specifically, there are 373 grids with observed annual rainfall erosivity values from 2014 to 2020. The $r_{\text{obs\_year}}$ and $r_{\text{ERA5\_year}}$ values in these grids were used to generate the multiplier factor map. The $r_{\text{obs\_year}}$ and $r_{\text{ERA5\_year}}$ values in other grids for 2014–2020, which were not used before, are available for

assessing the accuracy. Moreover, the year 2013 was regarded as a complete verification year, in which

the assessment of the $r_{\mathrm{cor\_year}}$ was conducted in all the TP grids with $r_{\mathrm{obs\_year}}$ values. Table 2 lists the

number of validation grids at each year, and Figure 3 shows the spatial distribution of the validation

grids for 2013–2020 (excluding 2017).

**Table 2. Numbers of grids used in this study**

| Year | Total number of grids with observations | Number of validation grids | Percentage of validation data in total data (%) |
|---|---|---|---|
| *2013* | *381* | *381* | *100* |
| 2014 | 477 | 104 | 22 |
| 2015 | 504 | 131 | 26 |
| 2016 | 562 | 189 | 34 |
| 2018 | 712 | 339 | 48 |
| 2019 | 745 | 372 | 50 |
| 2020 | 742 | 369 | 50 |

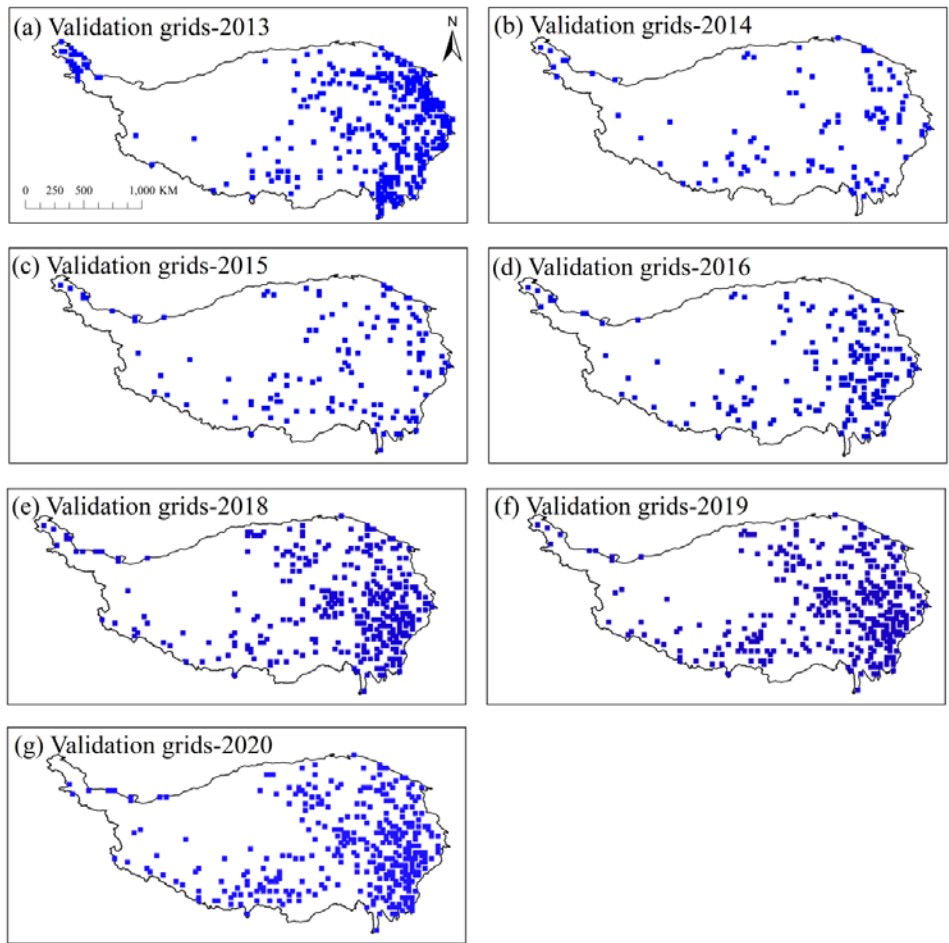

Figure 3. Spatial distribution of validation grids covering the TP for 2013–2020 (excluding 2017).

## 4 Results

### 4.1 Detecting accuracy of ERA5 for erosive rainfall events

Figure 4 compared the multi-year averaged annual erosive precipitation amount and frequency, and mean erosive event precipitation amount and $I_{30}$ derived from ERA5 precipitation data with those from in-situ observations. In detail, the EA5-based multi-year averaged annual erosive precipitation amount is three times more than the station-based value across the TP. The overestimation of the multi-year averaged annual precipitation amount was also reported by Jiao et al. (2021). The ERA5 overestimated the annual erosive precipitation frequency by 1.6 times. For the erosive event rainfall amount, ERA5 was almost twice as much as the station-based value, which differed from the finding of Jiao et al. (2021) that the daily precipitation amount with more than 10 mm are underestimated by ERA5. This result demonstrates that the erosive rainfall events in the TP cannot be simply equivalent to the daily

precipitation events (Chen et al., 2022). In addition, the mean $I_{30}$ of ERA5 for erosive events are only one ninth of the station-based value. Because the relatively slight overestimation of ERA5 precipitation data in the erosive event precipitation amount could not offset the substantial underestimation in $I_{30}$, the ERA5-based estimates showed a marked tendency to underestimate the rainfall erosivity when compared to the station-based estimates. Overall, the comparison between the two data sources indicated that there were significantly biases of ERA5 data in detecting precipitation characteristics for erosive events in the TP, however, also presented high corrections with correlation coefficient being 0.33–0.84.

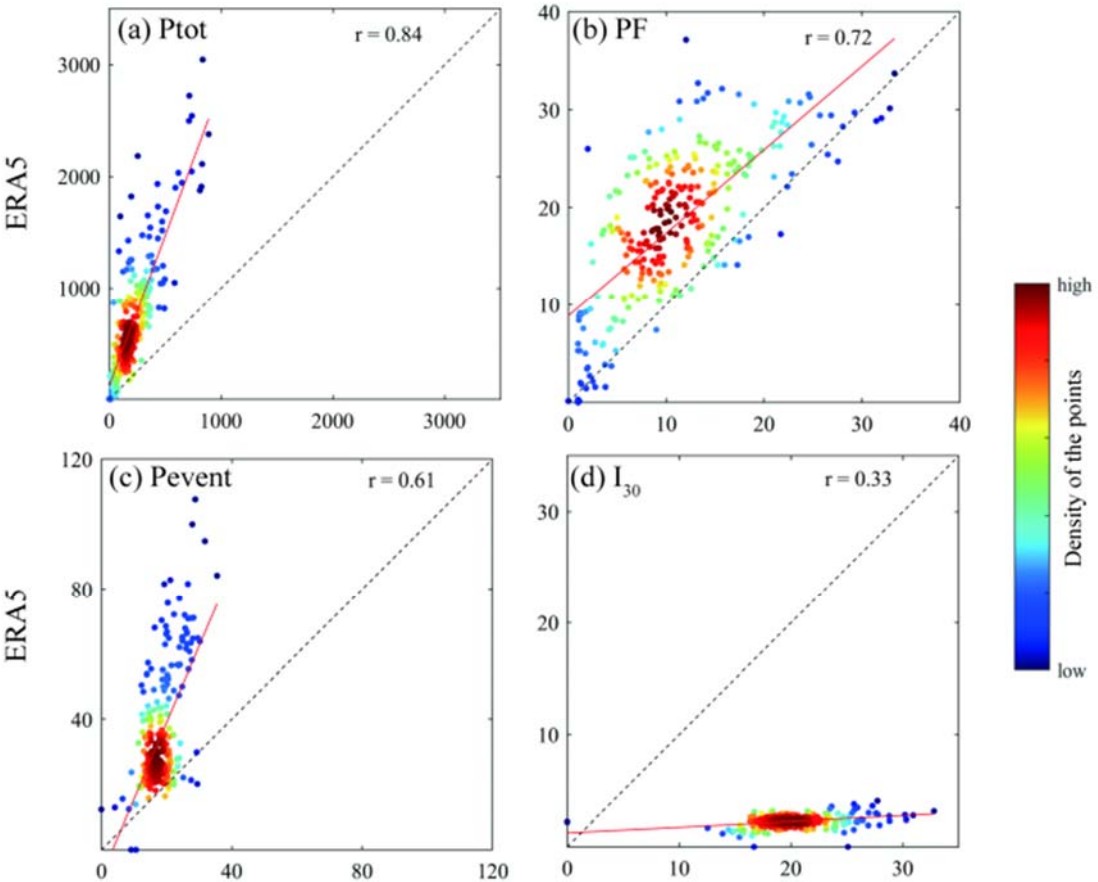

Figure 4. Scatterplots of the station-based multi-year averaged (a) annual erosive precipitation amount (Ptot: mm), (b) annual erosive precipitation frequency (PF: the number of annual erosive events), (c) mean erosive event precipitation amount (Pevent: mm), (d) mean $I_{30}$ for erosive events ( $I_{30}$: mm/h) vs those derived from ERA5 data at the corresponding grid cells in 2013–2020.

## 4.2 Evaluation of rainfall erosivity estimation using ERA5 data

The accuracy of annual rainfall erosivity estimation using the ERA5 precipitation data for 2013–2020 was assessed and compared with the $r_{obs\_year}$ values in 280 grids covering the TP. The correlation coefficient of the mean annual rainfall erosivity based on the observed and ERA5 precipitation data is 0.71. For most stations, the ERA5-based values were significantly underestimated (Figure 5).

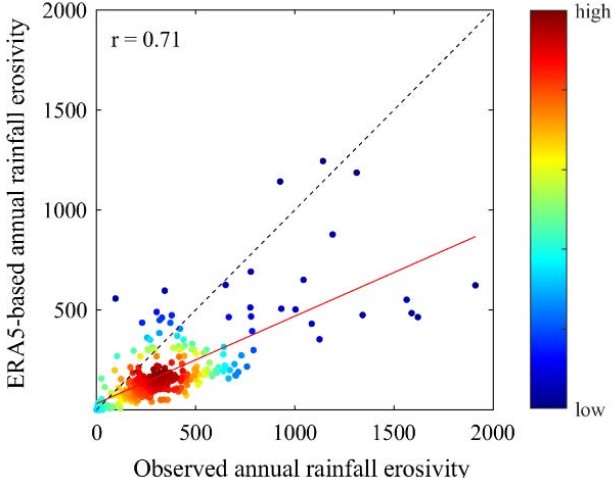

Figure 5. Comparison of mean annual rainfall erosivity based on observed and ERA5-based results for seven years (2013–2020, excluding 2017). The dotted line is the result of an optimal model (with an intercept of 0 and regression coefficient of 1). The red line is the regression result. Colors of dots represent the grid density. Unit: $MJ \cdot mm \cdot ha^{-1} \cdot h^{-1} \cdot yr^{-1}$.

To further evaluate the quality of mean annual rainfall erosivity estimation using ERA5 data, the performance of the ERA5 data in each grid was evaluated, as shown in Figure 6. The spatial pattern of the ERA5-based mean annual rainfall erosivity is consistent with that of the observed values. Specifically, areas with large annual rainfall erosivity are located mainly in the southeastern part of the plateau, especially at the southeast edge, whereas the mean annual values in the northwestern part of the plateau are relatively small. However, the observed mean annual rainfall erosivity on the TP is 344 $MJ \cdot mm \cdot ha^{-1} \cdot h^{-1} \cdot yr^{-1}$, and the ER5-based results underestimate this value by 47%. Moreover, except for most of the grids in the northwest corner and individual grids in the southeastern part of the plateau, the mean annual rainfall erosivity values in most grids in the TP are lower than the observed values.

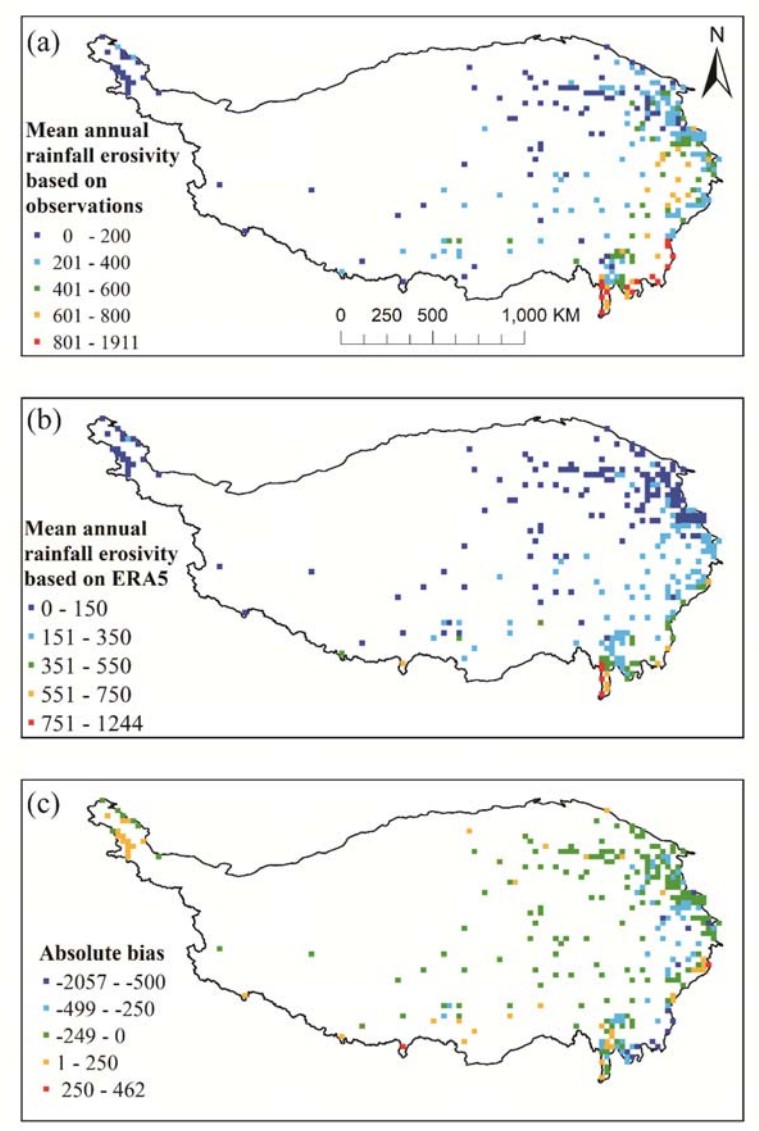

Figure 6. Mean annual rainfall erosivity in 2013–2020 (excluding 2017) based on (a) in situ precipitation observations and (b) ERA5 reanalysis precipitation data. (c) *AB* between the values based on ERA5 reanalysis data and precipitation observations. Unit: MJ·mm·ha$^{-1}$·h$^{-1}$·yr$^{-1}$.

The accuracy of the spatiotemporal variability of the mean annual rainfall erosivity on the TP obtained using the ERA5 dataset is also crucial for determining whether ERA5 is suitable for rainfall erosivity calculation. We used the first three EOF modes, which are considered to provide most of the valuable information, for evaluation. The spatial pattern of the first three EOFs of the observed values accounts for 77% of the total variance, and that of the first there EOFs of the ERA5-based values accounts for 84% of the total variance (Figure 7). Clearly, ERA5 successfully captured the spatial pattern of the EOF modes, especially the first two EOF modes, revealed by the observed values. In addition, the corresponding principal components of the EOF modes of the ERA5-based values are also consistent

with the temporal variation trend of the observed values. Therefore, it can be concluded that the

ERA5-based mean annual rainfall erosivity generally reproduces the spatiotemporal patterns of the

rainfall erosivity on the TP.

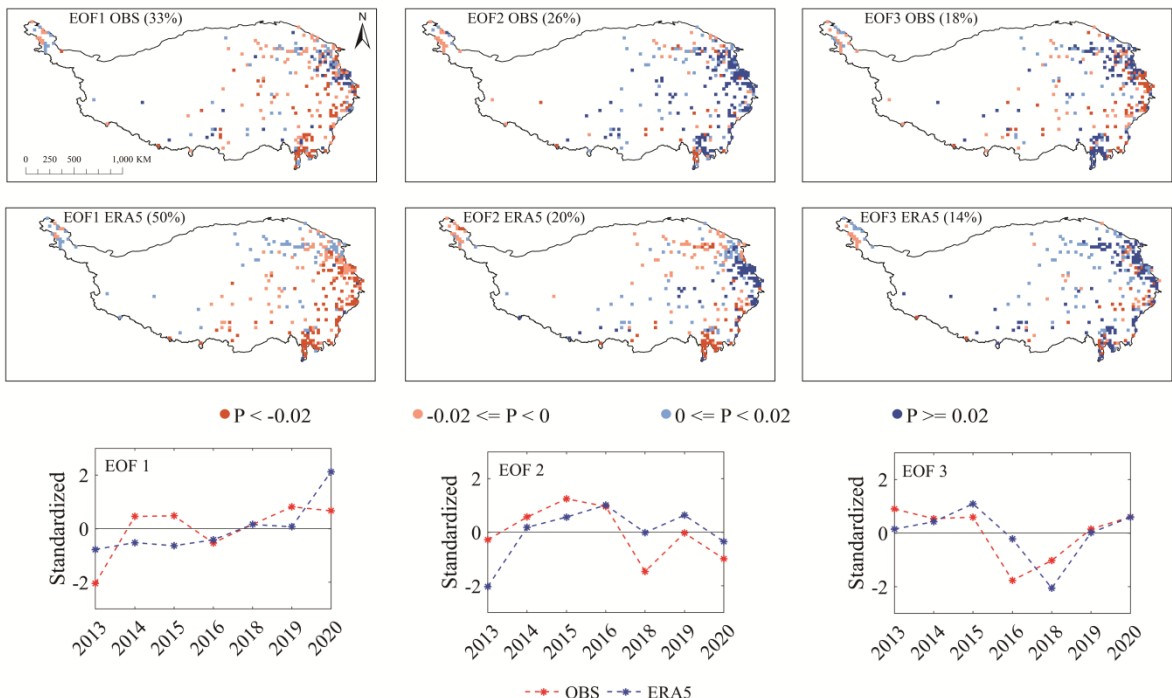

Figure 7. First three EOF modes of observed and ERA5-based mean annual rainfall erosivity on the TP in 2013–

2020 (excluding 2017).

## 4.2 Reconstruction and validation of corrected annual rainfall erosivity

Using the observed and ERA5-based annual rainfall erosivity, we calculated the multiplier factors for

373 grids [Figure 8(a)]. The multiplier factors for the TP range from 0 to 23, with a mean value of 2.4.

Multiplier factors of <1 indicate that the ERA5-based annual rainfall erosivity is overestimated, and

conversely, the annual rainfall erosivity in the grid is underestimated. Most of the areas with

overestimated ERA5-based mean annual rainfall erosivity are located in the Tarim, Qaidam, and

Yarlung Zangpo basins. In other areas, the annual rainfall erosivity is typically underestimated, and

areas with greater underestimation appear east of the Qaidam basin and in the source area of the Yellow

River. We also produced a multiplier factor map of the TP by IDW interpolation based on the multiplier

factors of 373 grids [Figure 8(b)].

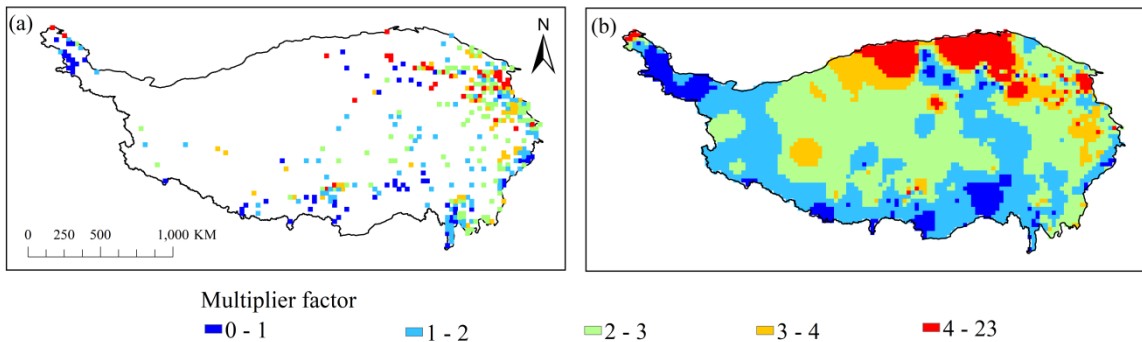

Figure 8. (a) Spatial distribution of multiplier factors of 373 grids, (b) multiplier factor map of TP generated by IDW interpolation.

The corrected annual rainfall erosivity in 2013–2020 (excluding 2017) was then calculated in the validation grids as the product of the ERA5-based annual values and multiplier factors from the map. Figure 9 compares the observed and ERA5-based annual rainfall erosivity in the validation grids by year. In 2014–2020 (excluding 2017), the multi-year averaged correction coefficient between $r_{\text{obs\_year}}$ and $r_{\text{cor\_year}}$ is 0.67, which is 0.13 larger than the value between $r_{\text{obs\_year}}$ and $r_{\text{ERA5\_year}}$. Moreover, all of the data for 2013, which were not used to produce the multiplier factor map, were used to conduct an independent assessment. The results show that the correction coefficient also increases, from 0.53 to 0.67, after the ERA5-based annual rainfall erosivity is corrected, indicating significant improvement.

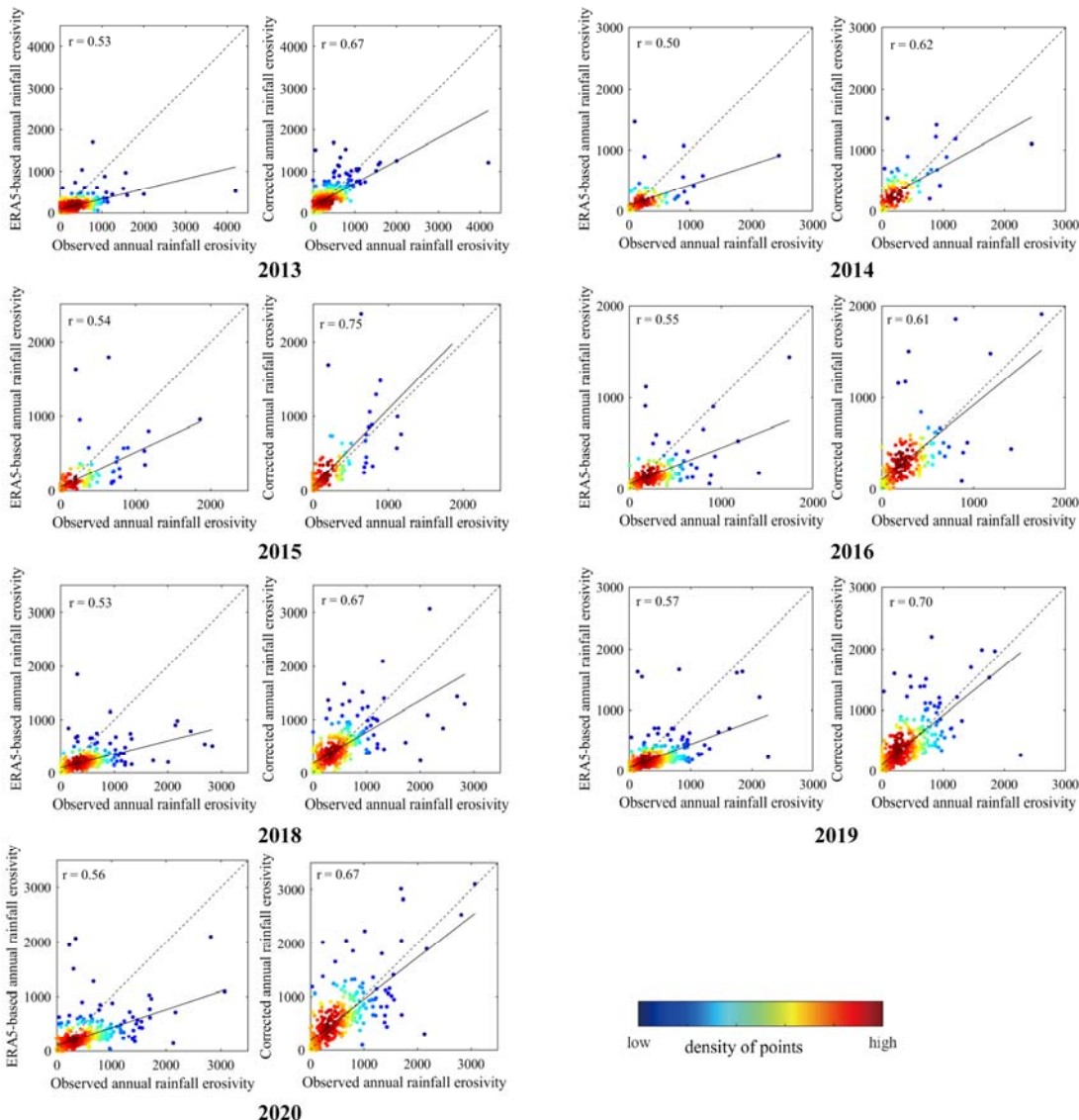

Figure 9. Comparison of ERA5-based annual rainfall erosivity (MJ·mm·ha⁻¹·h⁻¹·yr⁻¹) with observed values in validation grids for 2013–2020 (excluding 2017). The dotted line is the result of an optimal model (with an intercept of 0 and a regression coefficient of 1). The black solid lines are the regression result. Colors of dots represent the grid density.

Violin plots are an alternative method of synthetically evaluating the accuracy of the corrected annual rainfall erosivity. Figure 10 compares the observed, ERA5-based, and corrected annual rainfall erosivity in the validation grids for 2013–2020 (excluding 2017). The corrected annual rainfall erosivity values for 2014–2020 are better than the ERA5-based values in terms of both the probability density and the values corresponding to different quantiles. Even in 2013, a completely independent verification year, the accuracy of the corrected annual rainfall erosivity is greatly improved. Specifically, the observed grid-averaged multi-year mean annual rainfall erosivity is 329

MJ·mm·ha⁻¹·h⁻¹·yr⁻¹ in 2013–2020 (excluding 2017), where the ERA5-based value is 190

MJ·mm·ha⁻¹·h⁻¹·yr⁻¹, and the corrected value is 374 MJ·mm·ha⁻¹·h⁻¹·yr⁻¹. The relative error is

significantly reduced, from −42% to 14%, by multiplier factor correction.

355

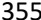

Figure 10. Violin plots of observed, ERA5-based, and corrected annual rainfall erosivity in validation grids for 2013–2020 (excluding 2017). *Y* axis shows annual rainfall erosivity in MJ·mm·ha⁻¹·h⁻¹. The boxplot diagram of the median of the violin plots shows the maximum value, 75% quantile value, 50% quantile value, 25% quantile value, and minimum value. The horizontal lines represent average values.

360

### 4.3 Rainfall erosivity in the TP and related uncertainties

Because of the large variability of the spatiotemporal patterns of precipitation, the $R$ factor, an essential input for soil loss estimation, must be calculated using a minimum of 20 years of precipitation data (Renard et al., 1997). In this study, the annual rainfall erosivity values of the TP for 71 years based on the 0.25° hourly ERA5 precipitation data were calculated by the algorithm shown in Section 3.1. Next, after correction by the multiplier factor map, the new annual rainfall erosivity dataset for 1950–2020 and $R$ factor map were produced.

The annual rainfall erosivity fluctuates considerably within a range of 239 to 408 MJ·mm·ha$^{-1}$·h$^{-1}$·yr$^{-1}$ (Figure 10). However, no obvious increasing or decreasing trend appears in the past 71 years across the TP. Regarding the spatial distribution, the $R$ factor generally shows a decreasing trend from southeast to northwest. The areas with $R$ factors below 200 MJ·mm·ha$^{-1}$·h$^{-1}$·yr$^{-1}$ are concentrated in the northwestern part of the TP, whereas regions with high $R$ factors appear mainly in the southeastern TP, especially in the Bomi–West Sichuan and Dawang–Chayu areas. The TP-averaged $R$ factor is 307 MJ·mm·ha$^{-1}$·h$^{-1}$·yr$^{-1}$, which is obviously lower than those from previous studies (e.g. Qin et al., 2016; Cao et al., 2018), excluding those of Liu et al. (2013) and Yue et al. (2022).

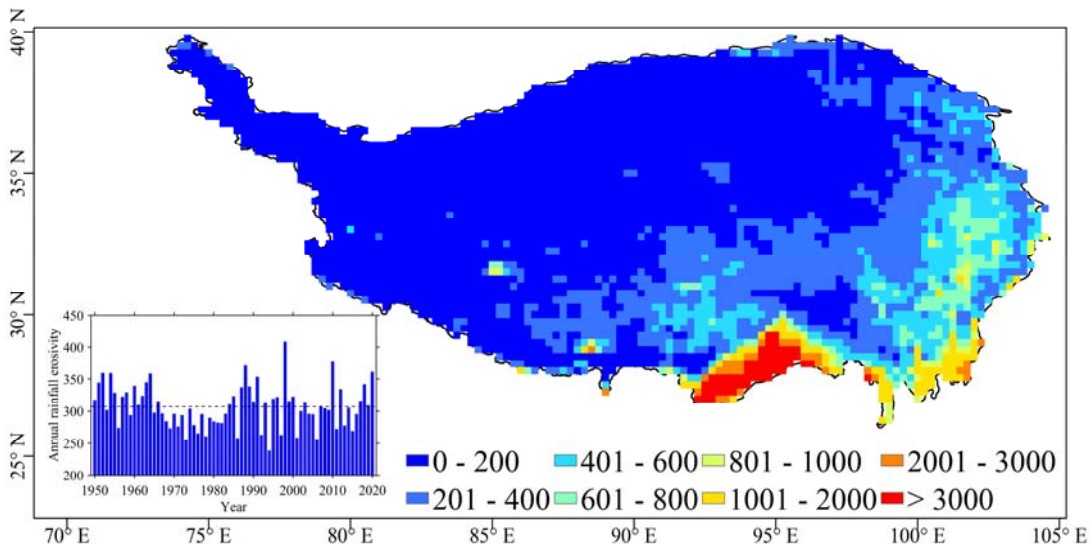

Figure 11. $R$ factor map of TP with the 0.25° spatial resolution for 1950–2020. Inset represents the yearly change in annual rainfall erosivity.

Compared with the previous studies, there are two essential improvements by using the data-driven approach in this study. One the one hand, the 1-min precipitation observations from 1787 weather stations are firstly used to calculate the accuracy rainfall erosivity values by employing the standard algorithm. With the densely-spaced rainfall erosivity values, it is able to yield realistic spatial distribution and identifying the high spatial heterogeneity of the rainfall erosivity over the TP. On the other hand, not only a $R$ factor map, we also produced a high-precision time series of annual rainfall erosivity for 71 years after correcting the ERA5-based estimations, which may offer great help to reveal the spatial-temporal evolution over the TP under the climate change.

It is also notable that some uncertainties are also unavoidably involved in the newly reconstructed dataset. As the biases of the ERA5 precipitation data in detecting the characteristics of the erosive rainfall events have been revealed, we intended to use multiple factors to correct the ERA5-based rainfall erosivity values by grid, to reduce the biases resulting from the ERA5 data. Limited by the scarcity of the in-situ precipitation observations from weather stations before 2013 (less than 100 weather stations), it is hardly to yield realistic spatial distribution of the multiple factor map by year. Here, we made a hypothesis that the biases of the ERA5-based annual rainfall erosivity always kept steady in various years, and thus the multi-year averaged annual multiple factor map from 2014–2020 is used in correcting process. With the improvement of the weather/climate forecast models in the future, the biases of the estimated rainfall erosivity by using gridded precipitation data will continue to reduce.

**5 Data availability**

The new gridded annual rainfall erosivity dataset for the TP for 1950–2020 is available at http://data.tpdc.ac.cn/en/data/37c34046-3c2a-4737-b3c9-35af398da62a/ (Chen et al., 2021).

**6 Conclusions**

This study presents a new gridded dataset of annual rainfall erosivity over the TP based on the 1-min in-situ precipitation data from 1787 weather stations and the long-term ERA5 precipitation data. The annual rainfall erosivity data are available over 71 years (from 1950 to 2020) on a 0.25° grid. The TP-averaged correction coefficient between the station-based annual rainfall erosivity and the newly released data is 0.67. In addition, the probability density and various quantile values of the new data are

generally consistent with the station-based values across the TP.

This dataset offers a unique view of large- to local-scale features in rainfall erosivity variability over the TP, where it is hardly to obtain the long-term precipitation data with sufficient spatial-temporal resolution. This new data availability opens up many interesting applications in soil erosion study and disaster research, including:

(1) providing input data of the $R$ factor, which is needed for soil erosion modelling;

(2) understanding the present processes of water erosion over the TP and improving future projections;

(3) identifying the hot spots at high risk of the landslide and flood hazards.

The data are available in Network Common (NC) Data Format that can be readily imported into standard geographical information system software (e.g. ArcGIS) or accessed programmatically (e.g. MATLAB, Python).

**Acknowledgments**

This research was jointly supported by the Second Tibetan Plateau Scientific Expedition and Research (STEP) Program (Grant No. 2019QZKK0307, Grant No. 2019QZKK0106), and the Basic Research Special Project of the Chinese Academy of Meteorological Sciences (Grant No. 2020Z003).

**Author contributions**

YC designed the study and wrote the manuscript. XD and MD contributed to the manuscript preparation and dataset generation. WQ, TW and JL contributed to the analysis method used in this study. YX contributed to the manuscript visualization.

**Declaration of interests**

The authors declare that they have no competing financial interests or personal relationships that could have influenced the work reported in this paper.

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
