# Peer review of "New gridded dataset of rainfall erosivity (1950–2020) on the Tibetan Plateau"

_Earth System Science Data, 2021_

## Author Comment (AC1)

The Tibetan Plateau (TP) is a hotpot for studying soil erosion under climate change. Rainfall erosivity (or the R factor) is the most widely used parameter regarding to climate in soil erosion study. Thus, an accurate R map and related dataset is benefit to quantify soil erosion on the TP. Generally, this is a good MS and provide valuable data on TP. I recommend a moderate revision. Please find bellow my suggestions for updating & reinforcing the current paper.

1.     Abstract: I suggest the authors add more information about the dataset.

**Response:** Thanks. As you suggested, we have rewritten the Abstract section to present a more clearer description of the reconstructed dataset, including the basic information **(Line 13 – 14)**, the reconstruction method **(Line 14 – 24)** and the performance of the newly released dataset **(Line 24 – 27)**.

2.     Introduction: The ERA5 should be introduce in this part. For example, I notice that this product has been used to calculate rainfall of the China's mainland.

**Response:** The introduction of the ERA5 dataset has been added. Please refer to **Line 100 – 106** for details in the revised MS.

3.     Line 112, Please check the spatial resolution of the ERA5 data, is 25 km or 0.25°?

**Response:** Thanks for your kind reminder. We have corrected the spatial resolution into 0.25°. Please refer to **Line 108** for details in the revised MS.

4.     Figure 4 and 5: Add the unit of rainfall erosivity, which is MJ·mm·ha−1·h−1·yr−1.

**Response:** Figure 4 and 5 have been modified to **Figure 5 and 6** in the revised MS, respectively. We have added the unit in the title of the figure.

5.      Part 4.2 (Line 243) Why the authors use multiplier factors to calculate new R map. Are there any references? Or the observed and ERA5-based annual rainfall erosivity show multiple relationship? Also in figure 8, why this is an optimal model with intercept of 0?

**Response:** Our study has found that ERA5 data has systematical biases in identifying the characteristics of the erosive precipitation events, including significant underestimation of the mean $I_{30}$ for erosive precipitation events and relatively slight overestimation of the mean erosive event precipitation amount, which will lead to overall underestimation of the ERA5-based annual rainfall erosivity.

In the post-processing of the precipitation simulation using weather/climate forecast models, it is always supposed that the model biases could keep stable, and consequently the relative changes between the in-situ and modeled precipitation are commonly used to correct the modeled precipitation for accuracy improvement (e.g. Fick et al., 2017; Cucchi et al., 2020; He et al., 2020). Here, taking the method of precipitation correction for a reference, we made the hypothesis that the biases of ERA5-based annual rainfall erosivity transmitted from the biases of ERA5 data can also keep stable at each grid. Then, the relative changes between the station-based and ERA5-based annual rainfall erosivity are used to correct the ERA5-based estimates. After the correction process, the performance of the corrected values is further examined. To make the method used in this study more clear, we have rewritten the section 3.3. Please refer to **Line 216－244** for details in the revised MS. Besides, the overall algorithm for generating the dataset has been illustrated in **Figure 2**.

In the revised MS, Figure 8 has been modified to Figure 9. In fact, the intercept of the optimal models is not equal to zero. We have changed the color of the fitting line to avoid misleading. Please refer to **Figure 9** for details in the revised MS.

**Reference**
Fick, S.E. and R.J. Hijmans: WorldClim 2: new 1km spatial resolution climate surfaces for global land areas. Int. J. Climatol., 37(12): 4302-4315, 2017.

Cucchi, M., Weedon, G. P., Amici, A., et al.: WFDE5: bias-adjusted ERA5 reanalysis data for impact studies, Earth Syst. Sci. Data, 12, 2097–2120, 2020.

He, J., Yang, K., Tang, W., et al: The first high-resolution meteorological forcing dataset for land process studies over China, Sci. Data., 7(1), 25, 2020.

6.      I suggest an additional part as Uncertainties in the results parts. Uncertainties either from the ERA5 or from the multiplier factors should be discussed.

**Response:** As you said, the biases of the ERA5-based annual rainfall erosivity data were derived from the ERA5 precipitation data, which has obvious biases in detecting the amount and intensity of erosive rainfall events. The correction method we used to reduce the biases unavoidably involved some uncertainties. Please see **Line 390 – 400** for the detailed analysis.

7.      Conclusions: I suggest this part focuses on summary of the dataset including its applications.

**Response:** As you suggested, we have rewritten the Conclusion Section in the revised MS. Please refer to **Line 406 – 411** for the summary of the dataset and **Line 412 – 419** for the application of the dataset.

---

## Author Comment (AC2)

The authors attempted to generate a new dataset of R factor on TP. This work is useful to for water erosion modelling. I recommend a major revision. Please find bellow my suggestions for the current paper.

1. Abstract: The spatial resolution of the generated gridded annual rainfall erosivity dataset should be added.

**Response:** Done. Please refer to **Line 17** in the revised manuscript (MS).

2. Introduction: I suggest the authors add more information of the previous studies about the *R* factor on the TP.

**Response:** As you suggested, the previous studies about *R* factor on the TP have been extended in the Introduction Section. First, we reviewed the previous studies, which employed the empirical methods to estimate the rainfall erosivity over the TP (**Line 74－84**). We found that the accuracy of the estimated rainfall erosivity in the TP are largely reduced by the current empirical estimation models and the scarcity of the historical weather stations. Second, the application of various gridded precipitation for rainfall erosivity estimation were reviewed (**Line 89－97**). We noted that significant biases of various gridded precipitation data have been identified in the TP, however, the gridded data for rainfall erosivity estimation are not prequalified, and the biases of estimated rainfall erosivity by using the gridded data have not been quantified and corrected. Besides, the previous studies have been summarized in **Table 1**.

3. Precipitation data: The detail information of the ERA5 data should be included in this section.

**Response:** Done. We have added the description of the ERA5 precipitation data in Line **146－150** in the revised MS.

4. The accuracy of the ERA5 data should be examined at the weather stations.

**Response:** We have analyzed the performance of the ERA5 precipitation data in the

terms of erosive precipitation characteristics over the TP, and found that ERA5 systematically overestimated the erosive annual amount, erosive annual precipitation frequency, and mean erosive event precipitation amount, but largely underestimated mean $I_{30}$ for erosive events. These biases of the ERA5 data in identifying the erosive precipitation events will transmit to the rainfall erosivity estimates over the TP. Therefore, it is necessary to correct the ERA5-based rainfall erosivity estimates to improve the accuracy. Please refer to **Section 4.1 (Line 253－269)** and **Figure 4** for details in the revised MS.

5. The procedure of how to reconstruction $R$ factor in 1950-2012 is important. However, no more information could be found in the methodology section. I would suggest the authors to rewrite this section to make this part more clearly.

**Response:** As you suggested, we have rewritten the Methodology Section. Generally, three steps are used to reconstruct the annual rainfall erosivity for the Tibetan Plateau with the 0.25° spatial resolution in 1950-2020 (**Line 159－164 and Figure 2**). We firstly calculated the annual rainfall erosivity by using the standard method of rainfall erosivity calculation based on the 1-min in-situ precipitation observations and 0.25° hourly ERA5 gridded precipitation data, respectively. Secondly, the performances of the ERA5 are systematically assessed in the terms of the detecting accuracy of the precipitation for erosive events and the estimation accuracy of ERA5-based annual rainfall erosivity. Thirdly, the annual rainfall erosivity data for the TP is generated by correcting the ERA5-based annual rainfall erosivity values.

Our study has found that ERA5 data has systematical biases in identifying the characteristic of the erosive precipitation events, including significant underestimation of the mean $I_{30}$ for erosive precipitation events and relatively slight overestimation of the mean erosive precipitation amount, which jointly lead to overall underestimation of annual rainfall erosivity estimates by using ERA5 data. Therefore, it is necessary to correct the biases of the ERA5-based annual rainfall erosivity estimates.

In the post-processing of the simulated precipitation using weather/climate forecast models, it is always supposed that the model biases could keep stable, and consequently the relative changes between the in-situ and modeled precipitation are commonly used to correct the modeled precipitation for accuracy improvement (e.g. Fick et al., 2017; Cucchi et al., 2020; He et al., 2020). Here, taking the method of precipitation correction for a reference, we make the hypothesis that the biases of ERA5-based annual rainfall erosivity transmitted from the ERA5 data can also keep stable at each grid. Then, the relative changes between the station-based and ERA5-based annual rainfall erosivity are used to correct the ERA5-based estimates. After the correction, the performance of the corrected values is further examined. To make the method used in this study more clearer, we have rewritten the **section 3.3**. Please refer to **Line 216 − 244** for details in the revised MS.

---

## Author Comment (AC3)

This paper developed a new long-term soil erosivity dataset using the ERA5 reanalysis precipitation product, corrected by 1-min time-step meteorological observations. The core of this work lies in the use of high-frequency precipitation measurements, which however were not fully presented and discussed. For example, what's the difference between this 1-min precipitation data and ERA-5 hourly product/other previous datasets? Furthermore, what are the contributions of precipitation frequency and intensity, respectively, to soil erosivity? These questions are important, considering the authors mentioned these issues in the introduction part. Overall, this paper does provide a valuable dataset for the TP region which is vulnerable to soil erosion but is generally superficial in clarifying the advantages and differences of this dataset against previous efforts. Therefore, I can only recommend a major revision for this paper in its current form.

**Response:** Thanks for your constructive suggestions. We have added the analysis of the difference between the 1-min in-situ precipitation data and ERA-5 hourly data ( **Line 253－269** and **Figure 4**). It is known that the rainfall erosivity is jointly determined by the erosive event rainfall amount and the maximum contiguous 30-min peak intensity. Our results have shown that the relatively slight overestimation of ERA5 in erosive event precipitation amount cannot offset the substantial underestimation of ERA5 in $I_{30}$, which jointly lead to the overall underestimation of the annual rainfall erosivity over the TP by using the ERA5 data.

On the other hand, we have further summarized the previous studies of the rainfall erosivity over the TP (**Line 74－97** and **Table 1**). Firstly, we reviewed the previous studies, which employed the empirical methods to estimate the rainfall erosivity over the TP (**Line 74－81**), and found that the accuracy of the estimated rainfall erosivity in the TP are largely reduced by the current empirical estimation models and the scarcity of the historical weather stations. Secondly, the application of various gridded precipitation for rainfall erosivity estimation were reviewed (**Line 89－95**). We have found that although the significant biases of various gridded precipitation data have been identified in the TP, however, the gridded data for rainfall erosivity estimation are not prequalified. The biases of rainfall erosivity estimates by using the gridded data

have not been quantified and corrected. In addition, limited by the scarcity of the long-term in-situ precipitation observation (< 30 weather stations before 1990, 30–100 weather stations in 1990–2012), the long time series of the rainfall erosivity over the TP is hardly to product.

Since 2012, China Meteorological Administration (CMA) has built a dense network of weather stations over the TP, and there are more than 1500 weather stations in and around the TP regions. Based on a vast number of in-situ precipitation data with high temporal resolution for 7 years, we can obtain the precise values of rainfall erosivity by using the standard method recommended by the USLE model. Meanwhile, we also use the same method to estimate the long-term annual rainfall erosivity by using ERA5 precipitation data. The new annual rainfall erosivity data is reconstructed by correcting the ERA5-based rainfall erosivity.

Considering that the systematical biases of ERA5-based estimates, we employ multiplier factor map in the correcting process of this study, which has been widely used to correct the simulated precipitation using weather/climate forecast models. The multiplier factor map is generated by IDW method based on the multi-year averaged relative changes between the station-based and ERA5-based annual rainfall erosivity. Here, we made the hypothesis that the biases of ERA5-based annual rainfall erosivity transmitted from ERA5 data can keep stable at each grid by year. After the new data has been produced, we further tested its accuracy. Overall, the long-term ERA5 precipitation data with systematical biases in detecting erosive precipitation events and the short-term in-situ observations with high-precise are jointly employed to reconstruct a long time series of annual erosivity in the TP. This data is of great importance to understand the spatial-temporal evolution of the rainfall erosivity over the TP. The advantages of the newly generated dataset against previous studies are described in **Line 382–389**.

**Minor comments**

Lines 307-309: The previous soil erosivity products are also based on observed precipitation data. I cannot see any difference of the methodology of this paper from the previous ones. Contrarily, the IDW method is actually too simple for the TP region, where the terrain is complex.

**Response:** In the previous studies, the multi-year average rainfall erosivity map ($R$ factor map) over the TP was directly interpolated by using IDW method based on the estimated rainfall erosivity values at dozens of weather stations. Given that the precipitation data from the weather stations with insufficient density, the released $R$ factor map over the TP cannot yield realistic spatial distributions of the rainfall erosivity. Yue et al (2022) has reported that the biases of $R$ factor over the TP was obviously larger than those in the other regions of China. Unlike the previous study, the IDW method in this study was used to generate the multiplier factor map, which was used to correct ERA5-based annual rainfall erosivity in this study.

Specifically, the multiplier factor values from 373 grids, which corresponded to appropriate 10% of the whole grids over the TP, were used to generate the multiplier factor map by IDW interpolation method. Notably, although the northwest region has scarce weather stations, the annual precipitation in these region is much less than other regions of the TP. In addition, the percentage of the grids with multiplier factor values in total grids was up to 15% in the hot spots of water erosion in the southern and eastern TP. On the southeast edge of the TP, the percentage of the available grid values were much higher.

We have tried our best to collect the latest precipitation data, especially the 1-min in-situ precipitation data from 1787 weather stations over the TP released by CMA, and we also believe that with the further development of the network of the weather stations, the accuracy of the rainfall erosivity estimates can be continuing to be improved.

**Reference**

Yue, T., Yin, S., Xie, Y., et al.: Rainfall erosivity mapping over mainland China based on high density hourly rainfall records, Earth Syst. Sci. Data., 14, 665-682, 2022.

---

## Author Response (AR2)

**1) The authors need to clarify why the author list has changed. What is the contribution of the new co-author?**

**Response:** There are six authors in the originally submitted version of the article. Because Yun Xie has provided some essential suggestions in the revision process, she was added as the seventh author. We also introduce her contribution in **Line 428** in the revised manuscript.

**2) In Figure 4, please add units for variables in each plot.**

**Response:** Done. Please refer to Figure 4 in the revised manuscript.